# Chaperone Networks in Fungal Pathogens of Humans

**DOI:** 10.3390/jof7030209

**Published:** 2021-03-12

**Authors:** Linda C. Horianopoulos, James W. Kronstad

**Affiliations:** Michael Smith Laboratories, Department of Microbiology and Immunology, University of British Columbia, Vancouver, BC V6T 1Z4, Canada; horianol@msl.ubc.ca

**Keywords:** heat shock proteins, chaperones, fungal pathogens, thermotolerance, antifungal drug tolerance

## Abstract

The heat shock proteins (HSPs) function as chaperones to facilitate proper folding and modification of proteins and are of particular importance when organisms are subjected to unfavourable conditions. The human fungal pathogens are subjected to such conditions within the context of infection as they are exposed to human body temperature as well as the host immune response. Herein, the roles of the major classes of HSPs are briefly reviewed and their known contributions in human fungal pathogens are described with a focus on *Candida albicans, Cryptococcus neoformans,* and *Aspergillus fumigatus*. The Hsp90s and Hsp70s in human fungal pathogens broadly contribute to thermotolerance, morphological changes required for virulence, and tolerance to antifungal drugs. There are also examples of J domain co-chaperones and small HSPs influencing the elaboration of virulence factors in human fungal pathogens. However, there are diverse members in these groups of chaperones and there is still much to be uncovered about their contributions to pathogenesis. These HSPs do not act in isolation, but rather they form a network with one another. Interactions between chaperones define their specific roles and enhance their protein folding capabilities. Recent efforts to characterize these HSP networks in human fungal pathogens have revealed that there are unique interactions relevant to these pathogens, particularly under stress conditions. The chaperone networks in the fungal pathogens are also emerging as key coordinators of pathogenesis and antifungal drug tolerance, suggesting that their disruption is a promising strategy for the development of antifungal therapy.

## 1. Introduction

Human fungal pathogens must be able to survive and proliferate in the host environment despite several unfavourable conditions. A major barrier to systemic infections in humans is the elevated mammalian body temperature compared to the ambient environment [1]. Human body temperature effectively restricts the growth of most fungi and, indeed, the majority of fungal infections are superficial and occur on the skin where temperatures are less restrictive [2,3]. Several human fungal pathogens also undergo morphological changes such as the transition between yeast and hyphae in response to temperature differences [4]. In order to survive at core human body temperature and to undergo these morphological changes, fungi must be able to cope with the proteotoxic stress induced at high temperature and upon flux in the demand for protein production. The heat shock proteins (HSPs) are one of the major groups of proteins which help respond to and mitigate these stresses. Herein, we will describe the general roles and classifications of HSPs, their known roles in the major human fungal pathogens, and the interactions between HSPs which coordinate chaperoning activity.

Classically, HSPs are defined as those proteins that are upregulated upon temperature upshift and stress induction as well as those which share a high degree of sequence similarity to established categories of HSPs [5]. Despite the general property of upregulation under stress conditions, there are also examples of HSPs that are constitutively expressed. Therefore, it is speculated that these proteins may have evolved to fulfill a proactive role of ensuring proper folding of nascent proteins and thus preventing the accumulation of proteotoxic stress rather than as a mechanism to respond to stresses such as heat shock [6]. The HSPs are involved in multiple processes including folding proteins de novo, stabilizing protein conformation under stress conditions, and modulating protein conformation to regulate their activity [7]. This modulation can involve individual proteins or multiprotein complexes which must be assembled or disassembled for their function [8].

HSPs have typically been named based on their sizes and classified based on their sequence similarity. This often reflects how they function, but provides limited information about their clients or the pathways in which they participate. In general, there are ATP-dependent HSPs including Hsp90, Hsp70, chaperonins, and disaggregases which undergo conformational changes upon ATP hydrolysis to facilitate protein folding, complex assembly, or disaggregation. There are also energy-independent passive chaperones such as the small HSPs which normally act as holdases to prevent protein aggregation. There are also many co-chaperones required to recruit clients as well as to facilitate the ATPase activity of their chaperones; however, we will focus on the Hsp40/J-domain co-chaperones of Hsp70s as they share a highly conserved domain and have emerging importance in fungal pathogens. These energy-dependent and -independent chaperones as well as their co-chaperones coordinately ensure that cells can function in normal and stressed conditions including those relevant to proliferation in a human host. The major classes of HSPs are reviewed below with descriptions of their major reported functions in the model species *Saccharomyces cerevisiae* to provide context, as well as current information on their known roles in the major human fungal pathogens (Figure 1).

## 2. Major Classes of HSPs

### 2.1. Hsp90s

Hsp90s are among the best studied HSPs due to the abundance of these proteins in cells. Hsp90 is an ATP-dependent chaperone which promotes substrate folding and participates in the activation of near native proteins through inducing conformational change [9,10]. Hsp90 has an N-terminal ATP-binding pocket, a C-terminal domain for homodimerization, and a C-terminal EEVD motif which promotes interactions with the tetratricopeptide repeat domains of several co-chaperones (Figure 2A) [11]. In *S. cerevisiae*, there are two paralogs encoding Hsp90 proteins and it is essential that cells have at least one functional copy. Hsp90s are among the most abundant proteins in the yeast cytosol under normal conditions [12]. One paralog (*HSC82*) is constitutively expressed and the other (*HSP82*) is induced upon heat shock; therefore, Hsp90 abundance is further increased upon heat shock [13]. Early reports on Hsp90s found that they interacted with steroid hormone receptors, kinases, actin, tubulin, calmodulin, and the SSA subfamily of Hsp70s [8,9,13,14,15]. Work using chemical genetic screens, synthetic genetic arrays, affinity purification, and yeast two hybrid analysis has expanded our understanding of the extensive nature of the Hsp90 interaction network. That is, Hsp90 interacts with approximately 10% of all yeast proteins, and was confirmed to interact with kinases, transcription factors, and other chaperones [16]. Chemical genetic screens under different conditions show that Hsp90 interacts genetically with the secretory pathway and cellular transport under normal growth conditions whereas Hsp90 interacted genetically with cell cycle, meiosis, and cytokinesis pathways upon elevated temperature (37 °C) [17]. More recently, proteomic approaches to identify interacting partners of the two paralogs encoding Hsp90 revealed that the interactomes of each protein are very similar in *S. cerevisiae* [18]. The clients that were identified reinforced previous studies which found that most client proteins were ligand-binding molecules. However, this approach revealed that most of these clients were enzymes involved in biosynthetic processes rather than kinases, as is the case with mammalian Hsp90 [18,19].

Despite the essentiality of Hsp90s in most fungal pathogens, several studies have been able to interrogate their contributions to pathogenesis using chemical inhibitors or strains with regulated Hsp90 expression. Early work using chemical inhibition established the importance of Hsp90 in the ability of fungi to gain resistance against antifungal drugs [20]. As expected for an essential protein, it was later shown that Hsp90 inhibition impairs fungal virulence [21] and influences the heat shock response through the Hsf1 transcription factor required for thermal adaptation after heat shock at 42 °C [22]. Additionally, combination therapy with anti-Hsp90 drugs and antifungal drugs of the azole class has broad therapeutic potential against multiple fungal pathogens [21,23]. In *C. albicans,* Hsp90 plays important roles in temperature-dependent morphological changes at 37 °C as well as in contributing to tolerance against the echinocandins [24,25]. Chemical genetic screens conducted under different conditions established the plasticity of the Hsp90 network in response to environmental stresses. These genetic screens also identified regulatory proteins which broadly interacted with Hsp90 under different conditions [26]. The physical interactions of Hsp90 and several of its co-chaperones have also been characterized in *C. albicans* revealing similarities with the interacting partners for *S. cerevisiae* Hsp90 under routine conditions, including clients participating in signal transduction. However, during exposure to antifungals, Hsp90 stabilizes P-bodies and stress granules thereby providing novel insights into the mechanisms by which Hsp90 promotes antifungal drug tolerance [27]. The roles of Hsp90 in *C. albicans* virulence have been reviewed in more detail elsewhere [28]. In *A. fumigatus*, Hsp90 also plays important roles in morphological transitions and has been suggested as a potential antifungal drug target in this context [29,30]. Furthermore, Hsp90 expression is upregulated upon cell wall stress in *A. fumigatus* and the kinases governing the cell wall integrity pathway such as MpkA and PkcA are clients of Hsp90 [31]. In *C. neoformans,* chemical inhibition of Hsp90 revealed its importance in capsule elaboration, thermotolerance at both 37 and 39 °C, tolerance to antifungal drugs, and virulence in a *Caenorhabditis elegans* model [32,33]. Overall, the importance of Hsp90 to fungal pathogenesis is well established making it one of the most promising candidates within the chaperone network as a target for antifungal therapies.

### 2.2. Hsp70s

Hsp70s have a “DnaK” domain, named after the *dnaK* gene first identified as necessary for bacteriophage λ DNA replication in *E. coli* [34]. Hsp70s promote folding of proteins through a cycle of substrate binding facilitated by co-chaperones and nucleotide exchange factors. The nucleotide binding domain of Hsp70 binds ATP, and co-chaperone-mediated ATP hydrolysis induces a conformational change in the substrate binding domain (SBD) closing the lid and allowing tight binding of client substrates (Figure 2B). Nucleotide exchange factors induce ADP release and ATP binding, which reverts the conformational change in the SBD to the open low affinity conformation ultimately resulting in substrate release [35,36]. Aside from this role in protein folding, Hsp70s participate in seemingly unrelated processes such as DNA replication, clathrin disassembly, and protein translocation across membranes, however these processes all require modulation of either protein conformation or protein-protein interactions [7,8,13]. Information from *S. cerevisiae* provides useful context for appreciating the roles of Hsp70 proteins. In this fungus, there are fourteen Hsp70 and Hsp70-like proteins in seven subfamilies including four typical subfamilies SSA, SSB, SSC, and KAR, and three atypical subfamilies, two of which function as nucleotide exchange factors (LHS, SSE) and one as a ribosome assembly protein (SSZ) [13,35,37]. There are also Hsp70 members in the endoplasmic reticulum (Kar2) and mitochondria (Ssc1), which are required for proteostasis within these organelles and are essential for viability [9,38]. Recent efforts have begun to elucidate the different roles of specific cytosolic Hsp70s. In *S. cerevisiae,* the Ssb1 and Ssb2 cytosolic Hsp70s associate with ribosomes and participate in the early folding of many nascent proteins [39]. Genetic studies also revealed that there are many genes encoding proteins that uniquely interact with each of the Hsp70s in the SSA subfamily [40]. Protein interaction studies show that the constitutive Hsp70s Ssa1 and Ssa2 interact with many more proteins than the inducible Ssa3 and Ssa4 proteins. However, this is based on information publicly available in the *Saccharomyces* genome database and it is likely that the full interactomes of Ssa3 and Ssa4 have not been characterized because the appropriate stress conditions have not been thoroughly tested [40]. Although the roles of Hsp70s have consistently been shown to relate to protein folding, the differences in interaction networks of cytosolic Hsp70s highlights their underappreciated specificity.

In the context of fungal pathogens, the Hsp70s have high levels of expression at elevated temperatures and also influence morphological transitions [41,42,43]. Hsp70s are found localized to the cell surfaces of *C. albicans* [44], *C. neoformans* [45], and *A. fumigatus* [46,47]. Accordingly, Hsp70s are immunogenic proteins in systemic infections caused by *C. neoformans* [48,49] or *C. albicans* [50]. The cytoplasmic Hsp70, Ssa1, contributes to virulence in *C. albicans* specifically through facilitating host cell invasion [51]. Furthermore, Ssa1 phosphorylation plays a role in regulating morphology and is required for thermotolerance at 42 °C in *C. albicans* [52]. In *C. neoformans,* Ssa1 contributes to melanin production in one serotype [53] and is required to promote non-protective M2 macrophage polarization in lung monocytes upon infection with a different serotype [54]; however, in both backgrounds, mutants lacking *SSA1* had reduced virulence in murine models of infection. Some Hsp70s are essential in fungal pathogens which makes it difficult to study their roles in virulence using classic reverse genetics approaches. However, the use of strains with regulated expression of the genes encoding these Hsp70s has revealed that their roles extend beyond known essential functions. For example, the Hsp70 protein Msi3 contributes to fluconazole resistance and virulence in *C. albicans* [55]. Similarly, the essential endoplasmic reticulum (ER) Hsp70, Kar2, is also crucial for thermotolerance at 37 °C as well as resistance against cell wall stress and azoles in *C. neoformans* [56]. Further investigations of the other members of the Hsp70 family and their roles in these human fungal pathogens may uncover their contributions to virulence.

### 2.3. Hsp40s/JDPs

The Hsp40s are co-chaperones necessary for stimulating the ATPase activity of Hsp70s and are thereby required for the Hsp70 cycle of client protein tight binding and release [57,58,59]. The Hsp40s are also called J domain proteins (JDPs) as they have a conserved ~70 amino acid J domain which is required for the stimulation of Hsp70 ATPase [59,60]. This name also comes from the DnaJ protein in *E. coli* which is encoded by a gene located next to *dnaK* and that is also required for bacteriophage λ DNA replication [61,62]. JDPs are classified into three categories based on the presence of conserved regions in addition to their J domains: type I JDPs have glycine/phenylalanine (G/F) rich domains and zinc binding motifs, type II JDPs have G/F rich domains, and type III JDPs only have a J domain [58,63,64]. The type I and II JDPs are able to bind non-native proteins through their G/F rich regions allowing them to bring these substrates to Hsp70s [58]. Most organisms have more JDPs than Hsp70s [65]. The expansion of this family contributes to the ability of JDPs to direct the activity of Hsp70s to a wide range of substrates and to participate in diverse functions. In particular, some cytosolic and ER-resident JDPs in *S. cerevisiae* are considered generalists based on their ability to bind a wide range of substrates, whereas some are specific for a small number of substrates [58,65]. Knockouts of generalist JDPs, such as Ydj1, can be rescued by overexpression of other JDPs; however, knockouts of specialist JDPs, such as Cwc23, Sis1, Jjj1, and Jjj3 are not rescued by overexpression of other JDPs [66]. Specialist JDPs can be specific for a certain substrate or process, as illustrated by the activity of Swa2 in clathrin-mediated endocytosis or Jac1 in iron-sulfur cluster assembly. Specificity of JDPs is largely governed by regions outside of the J domain and can be provided through recognition of conserved motifs such as polyglutamine stretches, or based on their position on or near ribosomes, mitochondria membranes, or the ER [65,66]. The JDPs which provide specificity based on their position can recruit Hsp70s, increase their local concentration, and direct the ATPase activity of these chaperones towards certain processes [64]. Many type III JDPs are poorly studied even in *S. cerevisiae* and it is generally thought that they participate in diverse and unique functions.

JDPs continue to be actively studied because many of their functions are still unknown even in model organisms. In human fungal pathogens, there are examples of JDPs which contribute to virulence mainly through participating in specific processes. For example, the cytosolic JDP, Ydj1, contributes to the stress response, morphogenesis, and mitochondrial function of *C. albicans* [67]. Furthermore, the mitochondrial JDP, Mrj1, contributes to mitochondrial respiration, capsule formation, and ultimately virulence in *C. neoformans* [68]. Surprisingly there are few other reports on the roles of JDPs in the virulence of human fungal pathogens despite their well characterized roles in fungal pathogens of insects [69,70] and plants [71,72,73,74,75]. Further study of JDPs, particularly the divergent type III JDPs, in the context of fungal pathogens may uncover novel and divergent proteins with roles in pathogenesis and also provide insights into the roles of their Hsp70 partners.

### 2.4. Hsp100s

The Hsp100/ClpB family is a group of hexameric AAA^+^ ATPase chaperones (Figure 2C) conserved in bacteria, yeasts, and plants but absent in metazoans [76,77]. These proteins generally function as disaggregases by pulling and processing protein strands from their client proteins through a central channel and unfolding them [77]. Paradoxically, they also increase prion propagation, likely by pulling polypeptide strands through their central pore and producing seeds for prion propagation [78]. In *S. cerevisiae*, the AAA^+^ ATPase Hsp104 is not essential for viability, but it is important for the acquisition of thermotolerance [79]. Importantly, members of the Hsp100/ClpB family also associate with other proteins such as proteases or other chaperone machinery [80]. This class of heat shock proteins also interacts with the Hsp70/JDP machinery, as discussed later. Therefore, it links together other aspects of the HSP machinery to coordinately mitigate proteotoxic stress.

In the context of human fungal pathogens, relatively little is known about the roles of Hsp100/ClpB in pathogenesis. However, in *C. albicans* Hsp104 contributes to biofilm formation as well as virulence in a worm model of infection [81]. Furthermore, the sumoylation of Hsp104 contributes to the thermotolerance to heat shock at 42 °C of *C. albicans* [82]. Although the roles of these chaperones have not been characterized in other fungi, Hsp104 is required for the acquisition of thermotolerance in *S. cerevisiae* [79] suggesting that it could potentially contribute to fungal adaptation to a mammalian host.

### 2.5. Chaperonins

Chaperonins are oligomeric protein folding complexes that have two stacked rings which allow unfolded proteins to enter the lumen. When an unfolded protein is interacting with the chamber, the chaperonin binds ATP and adopts a closed conformation creating a protected environment for proteins to fold (Figure 2D). Upon ATP hydrolysis, the chaperonins open and release the folded protein from the lumen [83]. The chaperonins are divided into two distinct groups. The group I chaperonins include the Hsp60s and GroEL, which are found in prokaryotes as well as in eukaryotic organelles such as mitochondria. The group II chaperonins, CCT/TRiC, are found in archaea and in the eukaryotic cytosol [84,85]. The mitochondrial chaperonin GroEL is involved in the folding of many mitochondrial proteins, especially those with a size less than 60 kDa and with regions of exposed hydrophobic β-sheets [86]. Early reports described the eukaryotic CCT chaperonin as being involved in the folding of actin and tubulin [84,87]. More recently, the roles of this chaperonin have been characterized in *S. cerevisiae* using proteomic and structural approaches. This analysis revealed that CCT participates in folding of proteins related to the nuclear pore complex, chromatin remodeling, protein degradation, the anaphase promoting complex, and the mTOR complex [88,89]. The CCT chaperonins are not as abundant as other chaperones, but they facilitate the folding of as many as 15% of all newly synthesized proteins [90].

The chaperonins play roles in morphogenesis and virulence in *C. albicans.* In particular, a CCT complex protein, Cct8, suppresses hyphal formation in *C. albicans* [91]. The CCT complex is also important for echinocandin resistance in *C. albicans* through its role in chaperoning actin dynamics, promoting correct septin localization, and maintaining cell wall architecture [92]. Sumoylation of Hsp60 is also required for regulation of hyphal growth in *C. albicans* and thermotolerance to heat shock at 42 °C when mitochondrial respiration is inhibited with antimycin A [82]. In addition to these studies, Hsp60 is immunogenic and has been proposed as a candidate vaccine target to protect against other human fungal pathogens including *Histoplasma capsulatum* and *Paracoccidioides brasiliensis* [93,94].

### 2.6. Small HSPs

Small HSPs (sHsps) are passive, energy-independent chaperones which oligomerize in part through their conserved α-crystallin domains. The monomeric proteins range in size from 12 to 40kDa and are often named accordingly (e.g., Hsp12) [95]. The sHsps demonstrate binding capacity for denatured substrates and they often act as holdases immobilizing denatured proteins until an ATP-dependent chaperone such as Hsp70 can bind and reactivate them [96]. In *S. cerevisiae*, Hsp12 is a well-studied sHsp whose expression increases with a variety of stresses including exposure to NaCl or ethanol [97]. Importantly, Hsp12 also plays a role in the plasticity and flexibility of the yeast cell wall [98].

The sHsps also contribute to fungal pathogenesis. For example, the sHsp Hsp21 is required for growth at elevated temperatures between 39 and 42 °C and ultimately virulence in *C. albicans* [99]. Hsp21 is also required for resistance against the antifungal drugs caspofungin and the imidazoles [100]. Another sHsp, Hsp12, is upregulated upon stresses including heat shock at 45 °C, oxidative stress, and osmotic stress, although it is not required for virulence in *C. albicans* [101]. In *C. neoformans,* the sHsps Hsp12 and Hsp122 contribute to resistance against amphotericin B [102]. Overall, these data suggest that sHsps may broadly be important for resistance against antifungal drugs. However, since the sHsps are diverse, the sHsps that are as yet uncharacterized may have novel contributions in the human fungal pathogens.

## 3. Coordination of Chaperoning Activity across Different Families of HSPs

The families of heat shock proteins reviewed here do not act in isolation. It has already been discussed how the JDPs are co-chaperones necessary for the ATPase activity of the Hsp70s. However, many of the HSP families act within chaperone networks to facilitate proper protein folding, complex assembly, and protein degradation. Within the Kingdom Fungi, the connections between HSPs are best characterized in *S. cerevisiae*, however many of these connections are not characterized in other fungi including the major human fungal pathogens (Figure 3A–D). In the STRING database, many of the connections between HSPs in the fungal pathogens are inferred from interactions between orthologs encoding HSPs in *S. cerevisiae.* Therefore, the interactions between the divergent proteins in these fungal pathogens largely remain unknown. In particular, many of the proteins without connections in the chaperone networks of human fungal pathogens were JDPs or sHSPs (Figure 3B–D). This suggests that there are divergent proteins in these classes of HSPs which may have specific functions related to pathogenesis and which warrant further research to assess their roles and potential as drug targets.

Perhaps the best characterized interactions between chaperones are between the Hsp70s and Hsp90s. In *S. cerevisiae,* Hsp90 and Hsp70 orthologs (Hsp82 and Ssa1) interact with each other mediated by the tetratricopeptide repeat protein Sti1 and this interaction is enhanced by cytosolic JDPs [15,103,104,105]. In addition to its role in coordinating their interaction, Sti1 facilitates substrate transfer from Hsp70 to Hsp90 [106]. Since Hsp90s are so abundant and participate in many pathways, efforts have been made to characterize their co-chaperones through establishing their genetic interaction network [107]. The Hsp90 co-chaperones generally participate in different processes as suggested by a lack of synthetic lethality between co-chaperone knockouts [107]. Characterization of the physical interactions of Hsp90s in *S. cerevisiae* also confirmed the interactions between Hsp90s and other chaperones and co-chaperones as well as the interactions with ligand binding biosynthetic enzymes [16,18,108].

The Hsp70 chaperone network is extensive and has been reviewed elsewhere [35] with particular focus on the roles of the JDPs in directing Hsp70 activity towards specific functions [65]. The JDPs are structurally diverse outside the J domain and often participate in discrete functions by directing the activity of Hsp70s through binding Hsp70 substrates, recruiting Hsp70s to a particular subcellular compartment, or modulating the Hsp70-substrate interaction [109]. The Hsp70s and JDPs also transfer substrates to other components of the chaperone machinery including the disaggregases and chaperonins [35,65,110]. The disaggregation of proteins in vitro is enhanced when both a type I and type II JDP such as Ydj1 and Sis1 are added to purified Hsp104, Ssa1, Sse1, and aggregated substrate compared to either JDP alone [111]. The Hsp70s and JDPs also transfer non-native substrates to the chamber-type chaperonins (mitochondrial GroEL and cytosolic T-complex chaperonin) where they can fold in a protected environment [35,65].

The chaperone network also connects the HSPs to other proteins and pathways relevant to protein homeostasis, most notably functions for protein degradation. The JDPs can direct substrates to particular fates by tethering chaperone machinery to specific subcellular compartments or to other proteins such as ubiquitin ligases for subsequent degradation [65]. The Sse subfamily of Hsp70s act as nucleotide exchange factors for other Hsp70s and facilitate the transfer of substrates from Hsp70 to the proteasome for degradation in *S. cerevisiae* [112]. Another class of chaperone, the T complex cytosolic chaperonin, also has several genetic interactions with protein degradation networks including proteins involved in ubiquitination and sumoylation [88]. These data for *S. cerevisiae* fit well with the interactome studies in mammalian cells which show that Hsp90 co-chaperone networks are extensive and connect to protein degradation through Hsp90′s interactions with ~31% of all E3 ligases [19,113]. The connectivity of the HSP network suggests that disruption of specific interactions can fine tune the chaperone network to modulate the suite of clients or chaperone activities while retaining other essential functions.

In the fungal pathogens of humans, the best characterized chaperone network is focused on Hsp90 and its co-chaperones in *C. albicans*. The genetic interaction network of Hsp90 in *C. albicans* under different conditions indicated that Hsp90 interacts with different proteins depending on the environmental stress conditions. However, two proteins, the kinase CK2, and the transcription factor Ahr1, interacted with Hsp90 in five of the six conditions tested. Indeed, CK2 and Ahr1 regulate the phosphorylation of Hsp90 and the transcription of the gene, respectively. Interestingly only ~ 17% of the genetic interaction network in *C. albicans* was shared with *S. cerevisiae* [26]. Physical interactions between chaperones in *C. albicans* also revealed that Hsp90 interacts with many other chaperones including Hsp70s and the T-complex chaperonins. However, this interactome characterization also revealed novel interactions with Hsp90 that had not been characterized in *S. cerevisiae* such as a connection with the osmotic stress responsive kinase Pbs2 [27]. Interactions between Hsp90 and Hsp70 via the StiA co-chaperone have also been demonstrated in *A. fumigatus* to coordinately promote tolerance to the antifungal drug caspofungin [114]. Together, these studies show both conserved and novel elements of the Hsp90 network in the human fungal pathogens. The novel elements discovered in *C. albicans* highlight the importance of characterizing the interactions between chaperones in the context of fungal pathogens. Similar studies in the other pathogenic fungi discussed in this review would likely uncover more novel elements of the chaperone network as both *C. albicans* and *S. cerevisiae* are within the class Saccharomycetes, whereas *A. fumigatus* is in a different class, Eurotiomycetes, and *C. neoformans* is even more distantly related as a basidiomycete. Furthermore, the pathogenic mechanisms of these fungi differ and hence the context-dependent interactions may also be novel in these fungal pathogens.

## 4. Heat Shock Proteins as Drug Targets for Fungal Pathogens

The importance of HSPs to virulence factor production in fungal pathogens as outlined above makes them attractive as potential targets for antifungal drug development [115]. Drugs which impair the functions of Hsp90 such as radicicol and geldanamycin have been used to study the functions and genetic interactions of Hsp90s in the context of human fungal pathogens [21,26,32,33]. Derivatives of geldanamycin such as 17-N-allylamino-17-demethoxygeldanamycin (17-AAG) which are less toxic to mammalian hosts have also shown promising antifungal activity [21,33,116]. Recently, fungal selective inhibitors of Hsp90 were developed which have activity against both *C. neoformans* and *C. albicans* Hsp90s [117,118]. The development of these inhibitors, which was guided by structural differences in the nucleotide binding domains between human and fungal Hsp90s complexed with inhibitors, allowed the generation of inhibitors with lower toxicity to human cells and therefore greater promise for clinical use [117,118].

Another approach to targeting the chaperone network is through disruption of protein-protein interactions between chaperones using small molecules. The impact of this approach differs from direct inhibition of HSP functions as their activities can be fine-tuned by disrupting interactions with a specific subset of co-chaperones or clients [119,120]. This approach has been discussed in the context of anti-cancer strategies although similar approaches could be used for antifungal drug development. In particular, this approach may allow targeting of the Hsp70 and co-chaperone network. Although, to our knowledge, these approaches have not been used or optimized for fungal pathogens, small molecules modulating the Hsp70/JDP interaction such as the dihydropyrimidines influence the phenotypes of a *ydj1*∆ in *S. cerevisiae* suggesting that they are bioavailable to fungi [121].

Thus far, many of the HSP inhibitors in use or in development target the well-conserved Hsp90s. The ability to make fungal selective inhibitors of Hsp90s is critical to ensuring that they can be used safely without off target effects. Studies characterizing chaperones contributing to virulence in fungal pathogens have identified divergent components of the chaperone network in fungi such as the sHSPs [99] and JDPs [68]. This information may be exploited to prioritize these divergent targets for antifungal drug development. Finally, there are still many uncharacterized chaperones and co-chaperones in the fungal pathogens of humans which are divergent from characterized chaperones in *S. cerevisiae.* Further characterization of these divergent proteins may identify novel targets for antifungal drug development in these pathogens.

## 5. Conclusions

The HSPs generally allow organisms to maintain proteostasis and survive despite exposure to high temperatures and other environmental stresses. The ability to survive elevated temperatures is one of the major natural factors limiting fungi from surviving and proliferating in human hosts. In addition to the general role of facilitating growth at human body temperature, the HSPs in fungal pathogens play important roles in promoting morphological changes associated with pathogenesis, resistance against antifungal drugs, and virulence in several infection models. Currently, inhibitors of the Hsp90s, which are crucial for fungal virulence, are being developed with improved selectivity for fungal proteins. As outlined in this review, almost every major class of HSPs has representatives which contribute to fungal pathogenesis. This includes HSPs which lack well-conserved proteins in humans. Therefore, there may be underexploited chaperone targets, and inhibition of these HSPs or disruption of the protein–protein interactions within the HSP networks are promising strategies to develop novel antifungal drugs and therapeutics.

## Figures and Tables

**Figure 1 jof-07-00209-f001:**
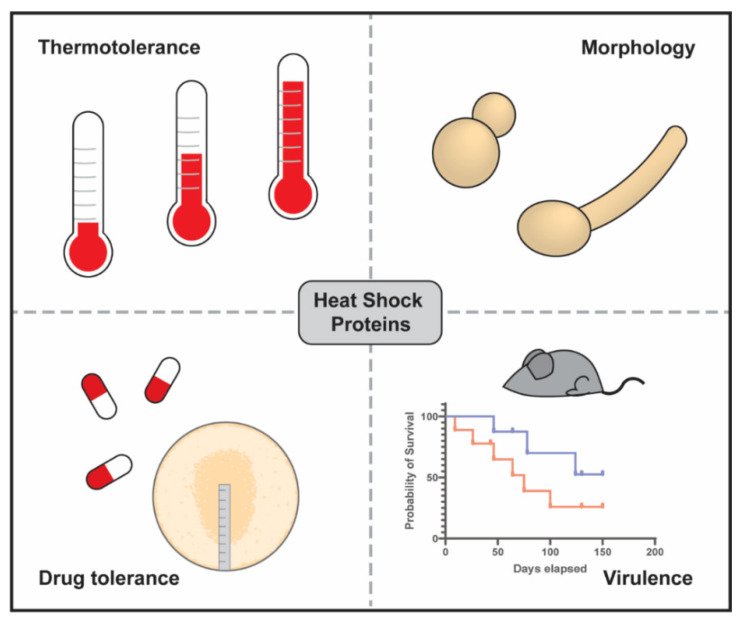
The major roles of heat shock proteins (HSPs) in fungal pathogens. The HSPs facilitate the acquisition of thermotolerance and allow human fungal pathogens to grow at human body temperature as well as to survive after heat shock at elevated temperatures. Several of these chaperones are required for morphological changes including the yeast to hyphal transition and conidiation. The HSPs, and in particular the Hsp90s, also facilitate antifungal drug tolerance in the major fungal pathogens suggesting that inhibitors would potentiate the activities of existing antifungals. Finally, in mammalian and insect models of infection when HSPs were pharmacologically inhibited or upon inoculation with deletion mutants lacking a gene for an HSP, they were often attenuated for virulence (indicated by the blue line in the hypothetical survival curve) compared to wild type or untreated strains (red line).

**Figure 2 jof-07-00209-f002:**
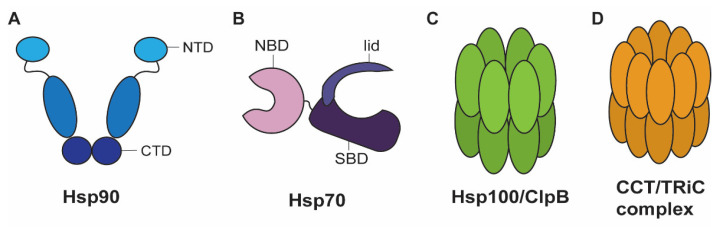
Schematic diagrams of the major classes of chaperones. (**A**) An Hsp90 dimer schematic indicating the ATP binding N-terminal domain (NTD) and the C-terminal domain (CTD) which is required for homodimerization. (**B**) An Hsp70 schematic indicating the nucleotide binding domain (NBD) and the substrate binding domain (SBD) including the lid of the SBD in an open conformation. (**C**) A schematic of an Hsp100/ClpB chaperone showing the stacked hexameric rings through which peptide strands are pulled to unfold them. (**D**) A schematic of a CCT/TRiC complex showing the stacked oligomeric rings which form a protected environment for protein folding in the lumen when ATP is bound and the chaperonin adopts a closed conformation.

**Figure 3 jof-07-00209-f003:**
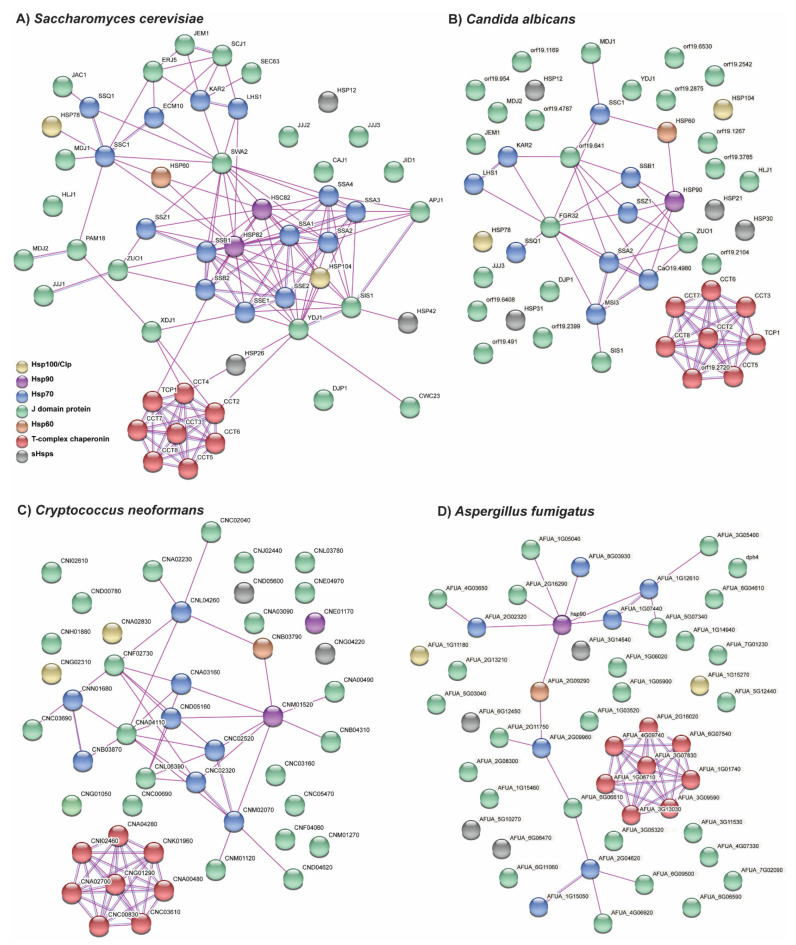
Network analyses of known and predicted interactions between chaperones in (**A**) *S. cerevisiae,* (**B**) *C. albicans,* (**C**) *C. neoformans,* and (**D**) *A. fumigatus.* STRING network analyses (https://string-db.org/, accessed on 5 January 2021) displaying the interactions between members of the Hsp100, Hsp90, Hsp70, JDP, sHsp, and chaperonin (Hsp60 and T-complex) families in *S. cerevisiae* and three of the major human fungal pathogens. Only experimentally determined interactions are shown including those inferred from interactions between putative homologs in other species. There are few known interactions with Hsp100s, JDPs, and sHsps even in *S. cerevisiae.* Furthermore, there are fewer characterized interactions within the chaperone networks of the human fungal pathogens compared to *S. cerevisiae.* Gene names are given where known and gene identification numbers from FungiDB (https://fungidb.org/fungidb/app, accessed on 5 January 2021) are provided otherwise (from *Candida albicans* SC5314, *Cryptococcus neoformans* var. *neoformans* JEC21, and *Aspergillus fumigatus* Af293).

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
