# Peer review of "Chaperone Networks in Fungal Pathogens of Humans"

_jof, 2021, doi:10.3390/jof7030209_

Round 1

Reviewer 1 Report

The paper by Linda C. Horianopoulos and James W. Kronstad is a mini-reviews of some data about fungal chaperones. Main problem is that it provides too poor information about fungi-specific features of chaperone network and structure of chaperones. Such information is of special importance if authors consider chaperones as putative targets for antifungal therapy. It is obvious that essential proteins can be considered as drug targets, but a key problem is that such proteins should differ enough from human homologs.

Other comments:

The paper is focused on Candida albicans, Cryptococcus neoformans, and Aspergillus fumigatus. Why these pathogens? Second, some comments should be added on the taxonomy of these pathogens as well as all other species mentioned in the text: are they close relative or not and (if possible) do they have some specific features or not.

In discussions about HSPs in terms of heat shock and thermotolerance, higher temperature is usually mentioned, not 37° (for example, 50-60° for GroEL/GroES). So, additional comments should be added to clarify this point.

Section 4 – I suggest to add more information about structural difference between fungal and human chaperones and strategies to use this difference for developing anti-fungal drugs.

Author Response

Review 1:

Comments and Suggestions for Authors

The paper by Linda C. Horianopoulos and James W. Kronstad is a mini-reviews of some data about fungal chaperones. Main problem is that it provides too poor information about fungi-specific features of chaperone network and structure of chaperones. Such information is of special importance if authors consider chaperones as putative targets for antifungal therapy. It is obvious that essential proteins can be considered as drug targets, but a key problem is that such proteins should differ enough from human homologs.

We appreciate the comment about providing more information on fungal specific features of the chaperone network. We have made a few changes to highlight the divergent chaperones and co-chaperones in fungal pathogens of humans as potential targets for antifungal drug development. Unfortunately, there is still a lack of information about divergent chaperones in fungal pathogens aside from some of the recent publications we discussed. It is our hope that this review will highlight these missing links and prompt further work to address these important questions.

In section 3 we have added lines 319-323: “In particular, many of the unconnected proteins in the chaperone networks of human fungal pathogens were JDPs or sHSPs (Fig. 2B-D). This suggests that there are divergent proteins in these classes of HSPs which may have specific functions related to pathogenesis and warrant further research to assess their roles and potential as drug targets.”

In section 3, we have also rearranged the information and put the fungal pathogen specific information in a separate paragraph (lines 372-394)  for clarity and added a few sentences on the potential of finding novel elements in the chaperone networks of A. fumigatus and C. neoformans using the studies on Hsp90 in C. albicans as a model.

We have also condensed the information about the Hsp70 and Hsp40 chaperones in S. cerevisiae so that these sections are more balanced between background information and fungal pathogen specific information.

Other comments:

The paper is focused on Candida albicansCryptococcus neoformans, and Aspergillus fumigatus. Why these pathogens? Second, some comments should be added on the taxonomy of these pathogens as well as all other species mentioned in the text: are they close relative or not and (if possible) do they have some specific features or not.

These pathogens were chosen as they are the major pathogens responsible for systemic fungal infections of humans in which HSPs are presumed to be more important.  Also, there is generally more information available on the HSPs in these pathogens.

In our speculation that there may be novel elements of the HSP network in A. fumigatus and C. neoformans, we have included the following information on their evolutionary relationships (lines 389-394):

“Similar studies in the other pathogenic fungi discussed in this review would likely uncover more novel elements of the chaperone network as both C. albicans and S. cerevisiae are within the class Saccharomycetes, whereas A. fumigatus is more distantly related in the class Eurotiomycetes and C. neoformans is even more distantly related as a basidiomycete. Furthermore, the pathogenic mechanisms of these fungi differ and hence the context-dependent interactions may also be novel in these fungal pathogens.”

In discussions about HSPs in terms of heat shock and thermotolerance, higher temperature is usually mentioned, not 37° (for example, 50-60° for GroEL/GroES). So, additional comments should be added to clarify this point.

Thank you for bringing this to our attention, we generally focused on temperatures more relevant to proliferation of fungi in human hosts rather than these higher temperatures. For clarity, we have added the temperatures that studies were done at (for example, 37°C growth or 42°C heat shock) where relevant. Please also see our response to a similar question from Reviewer 2 regarding A. fumigatus and higher temperatures.

Section 4 – I suggest to add more information about structural difference between fungal and human chaperones and strategies to use this difference for developing anti-fungal drugs.

To address this issue, we have added more specific information about the structurally-guided approaches to develop fungal selective inhibitors against Hsp90, as follows lines 404-407 “The development of these inhibitors, which was guided by structural differences in the nucleotide binding domains between human and fungal Hsp90s complexed with inhibitors, allowed the generation of inhibitors with lower toxicity to human cells and therefore greater promise for clinical use [117,118].”

To our knowledge similar studies have not been conducted on other HSP’s. Therefore, we have suggested that recently identified divergent HSPs may make promising targets for antifungal drug development as an alternative to targeting structural differences in conserved proteins.

Reviewer 2 Report

This is a very complete and dense review on chaperones in fungal pathogens. It is generally well written. It brings a lot of information and the reviewer learned a lot reading it. Nevertheless, it might be easier to read if some paragraphs were more focused on fungal pathogens. Moreover, a few modifications, mostly of the structure of the manuscript could improve its quality.

  • In the introduction, it might be interesting to have a general presentation of the different classes of chaperones and co-chaperones, which will further detailed in the review. It would be also interesting to define right away, what are the chaperones and the co-chaperones.
  • In that sense, whereas it is very clear that Hsp70s, CCTs have chaperone activities, experiments demonstrating Hsp90 chaperone activity are not presented. Are Hsp90 co-chaperones or chaperones? In this review, it might be better to first present the chaperones and then the co-chaperones.
  • A schematics showing the structure and the domains of each types of proteins will help the readers to follow the structural description of the proteins.
  • In the paragraph Hsp70s, although interesting the part on the general knowledge on Hsp70s (what is known is S. cerevisiae) is too long as compared to the fungal pathogen part, which is the theme of the review. This first part of the paragraph should shorten and simplified.
  • Same remark for the Hsp40 paragraph.
  • The tittle of the part 3 is poorly chosen as this proteostasis is just described in the last paragraph
  • The figure 2 should be limited to S. cerevisiae as the other predictions of interactions take into account the cerevisiae network to be constructed. It suggest a putative conservation of the network whereas most of the experiments have been done in only one species.
  • A general question, which might be develop in the conclusions and perspectives paragraph: Some organisms like A. fumigatus are able to grow at very high temperature (more than 50°C). It there some published descriptions of specifies of the HSP network in this fungal pathogen which might explain this phenotype?

Minor

  • Introduction: the reviewers is somehow surprised to read that fungal pathogens affect mostly the extremities.
  • It the paragraph on Hsp90, line 100. “It was further shown Hsp90 inhibition impairs fungal virulence”. It is not expected knowing that it is essential? Something like “ As expected,  Hsp90…” would be better.

Author Response

Review 2:

This is a very complete and dense review on chaperones in fungal pathogens. It is generally well written. It brings a lot of information and the reviewer learned a lot reading it. Nevertheless, it might be easier to read if some paragraphs were more focused on fungal pathogens. Moreover, a few modifications, mostly of the structure of the manuscript could improve its quality.

In the introduction, it might be interesting to have a general presentation of the different classes of chaperones and co-chaperones, which will further detailed in the review. It would be also interesting to define right away, what are the chaperones and the co-chaperones.

We agree that this point may give readers more information about what to expect from this review, therefore we have added the following, lines 57-67:

“In general, there are ATP-dependent HSPs including Hsp90, Hsp70, chaperonins, and disaggregases which undergo conformational changes upon ATP hydrolysis to facilitate protein folding, complex assembly, or disaggregation. There are also energy-independent passive chaperones such as the small HSPs which normally act as holdases to prevent protein aggregation. There are also many co-chaperones required to recruit clients as well as to facilitate the ATPase activity of their chaperones, however we will focus on the Hsp40/J-domain co-chaperones of Hsp70s as they share sequence similarity and have emerging importance in fungal pathogens. These energy-dependent and -independent chaperones as well as their co-chaperones coordinately ensure that cells can function in normal and stressed conditions including those relevant to proliferation in a human host.”

In that sense, whereas it is very clear that Hsp70s, CCTs have chaperone activities, experiments demonstrating Hsp90 chaperone activity are not presented. Are Hsp90 co-chaperones or chaperones? In this review, it might be better to first present the chaperones and then the co-chaperones.

Thank you for bringing this lack of clarity to our attention. We hope that the addition to the introduction will help to clarify this. To further clarify the role of Hsp90 as a chaperone, we have added the following to the section on Hsp90’s lines 84-85:

“Hsp90 is an ATP-dependent chaperone which promotes substrate folding and participates in the activation of near native proteins through inducing conformational change [9,10].”

A schematics showing the structure and the domains of each types of proteins will help the readers to follow the structural description of the proteins.

We appreciate this suggestion and have made the following figure. We have provided schematics for the structurally conserved chaperones, Hsp90, Hsp70, Hsp100/ClpB, and CCT/TRiC complex. The JDPs and sHSPs were not included as they are structurally diverse.

Figure 2. Schematic diagrams of the major classes of chaperones. (A) An Hsp90 dimer schematic indicating the ATP binding N-terminal domain (NTD) and the C-terminal domain (CTD) which is required for homodimerization. (B) An Hsp70 schematic indicating the nucleotide binding domain (NBD) and the substrate binding domain (SBD) including the lid of the SBD in an open conformation. (C) A schematic of an Hsp100/ClpB chaperone showing the stacked hexameric rings through which peptide strands are pulled through to unfold them. (D) A schematic of a CCT/TRiC complex showing the stacked oligomeric rings which form a protected environment for protein folding in the lumen when ATP is bound and the chaperonin adopts a closed conformation.

In the paragraph Hsp70s, although interesting the part on the general knowledge on Hsp70s (what is known is S. cerevisiae) is too long as compared to the fungal pathogen part, which is the theme of the review. This first part of the paragraph should shorten and simplified.

Same remark for the Hsp40 paragraph.

We have simplified these introductory paragraphs to focus on the major functions and necessary information to understand the roles in fungal pathogens. We also included text to make it clear why the information from S. cerevisiae provides useful context for understanding the roles of the HSPs in the fungal pathogens.

The tittle of the part 3 is poorly chosen as this proteostasis is just described in the last paragraph

We have changed the title of this section so it reflects our focus on coordination of activities between HSPs of different families. The new title is: “Coordination of chaperoning activity across different families of HSPs”

The figure 2 should be limited to S. cerevisiae as the other predictions of interactions take into account the cerevisiae network to be constructed. It suggest a putative conservation of the network whereas most of the experiments have been done in only one species.

We appreciate the reviewer’s suggestion, but our goal with this figure is to highlight the nodes in the fungal pathogens which are not connected to the network. It is our intention to explore  the idea that there are elements of the HSP network which are divergent and may be interesting to follow up on as potential targets for antifungal drug development. We have added the following sentences to emphasize this point lines 319-323:

“In particular, many of the unconnected proteins in the chaperone networks of human fungal pathogens were JDPs or sHSPs (Fig. 3B-D). This suggests that there are divergent proteins in these classes of HSPs which may have specific functions related to pathogenesis and warrant further research to assess their roles and potential as drug targets.”

A general question, which might be develop in the conclusions and perspectives paragraph: Some organisms like A. fumigatus are able to grow at very high temperature (more than 50°C). It there some published descriptions of specifies of the HSP network in this fungal pathogen which might explain this phenotype?

This is a very interesting question. Overall, we think that this highlights the importance of doing these studies in the context of these fungal pathogens and with conditions such as these to identify novel elements of the chaperone network and their importance to the pathogenesis of that fungus. We note that Albrecht et al 2010 found that in response to heat shock at 48°C, the HSPs increased in abundance in A. fumigatus similarly to how HSPs are upregulated in S. cerevisiae. However, a few novel targets of the transcription factor Hsf1 were found such as the nuclear migration protein NudC, the mannitol biosynthesis protein mannitol-1-phosphate dehydrogenase, and the reactive oxygen intermediate cytochrome c peroxidase. Since this information is not directly related to the chaperone network, we have not included it in this review although it is important to consider in the context of thermotolerance.

Minor

Introduction: the reviewers is somehow surprised to read that fungal pathogens affect mostly the extremities.

We have clarified this point to highlight that these occur on the skin. This sentence now reads (lines 32-34):

“Human body temperature effectively restricts the growth of most fungi and, indeed, the majority of fungal infections are superficial and occur on the skin where temperatures are less restrictive.”

It the paragraph on Hsp90, line 100. “It was further shown Hsp90 inhibition impairs fungal virulence”. It is not expected knowing that it is essential? Something like “ As expected,  Hsp90…” would be better.

We have fixed this and it now reads (lines 125-127):

“As expected for an essential protein, it was later shown that Hsp90 inhibition impairs fungal virulence [21] and broadly influences the heat shock response through the Hsf1 protein required for thermal adaptation [22].”